# Nuclear phase retrieval spectroscopy using resonant x-ray scattering

Ziyang Yuan [1,2,3,4,12], Hongxia Wang[4,12], Zhiwei Li [5], Tao Wang[5], Hui Wang[5], Xinchao Huang[6], Tianjun Li[6], Ziru Ma[6], Linfan Zhu[6], Wei Xu [7], Yujun Zhang [7], Yu Chen[7], Ryo Masuda[8], Yoshitaka Yoda [9], Jianmin Yuan[10], Adriana Pálffy [11] ✉ & Xiangjin Kong [1,2] ✉

Light-matter interaction is exploited in spectroscopic techniques to access information about molecular, atomic or nuclear constituents of a sample. While scattered light carries both amplitude and phase information of the electromagnetic field, the latter is lost in intensity measurements. However, often the phase information is paramount to reconstruct the desired information of the target, as it is well known from coherent x-ray imaging. Here we introduce a phase retrieval method which allows us to reconstruct the field phase information from two-dimensional time- and energy-resolved spectra. We apply this method to the case of x-ray scattering off Mössbauer nuclei at a synchrotron radiation source. Knowledge of the phase allows also for the reconstruction of energy spectra from two-dimensional experimental data sets with excellent precision, without theoretical modelling of the sample. Our approach provides an efficient and accurate data analysis tool which will benefit x-ray quantum optics and Mössbauer spectroscopy with synchrotron radiation alike.

X-ray scattering is a powerful tool for imaging or even ghost imaging[1,2], given the involved wavelengths which are commensurate with molecular or interatomic distances[3,4]. Resonant x-ray scattering on the other hand often involves atomic transitions of core electrons or narrow resonances of Mössbauer nuclei[3]. The latter occur at x-ray wavelengths and can be considered as ideal quantum systems with high quality factors[5]. For instance, the most widely used Mössbauer nuclear resonance is the transition between the ground state and the first excited state of $^{57}$Fe, for which the energy resolution $\Delta E/E$, defined as the ratio of transition linewidth $\Delta E$ and resonant energy $E$, is $3 \times 10^{-13}$.

This energy resolution is even higher for other Mössbauer nuclei, like $^{45}$Sc ($\Delta E/E \sim 10^{-19}$)[6] and $^{103}$Rh ($\Delta E/E \sim 10^{-24}$)[7]. Such nuclear resonances can be very sensitive to their environment, providing sensitive information in various fields such as physics, chemistry, biology or metallurgy. Mössbauer nuclei have been used to determine the gravitational redshift[8–10], to study magnetically ordered materials[11] and to investigate the mineralogy of iron-bearing rocks and soils[12]. Meanwhile, with the development of x-ray sources and detecting devices, Mössbauer nuclei of exceptionally narrow resonances are treated as a promising platform to implement quantum optics or coherent control in the hard

¹Key Laboratory of Nuclear Physics and Ion-Beam Application (MOE), Institute of Modern Physics, Fudan University, Shanghai 200433, China. ²Research Center for Theoretical Nuclear Physics, NSFC and Fudan University, Shanghai 200438, China. ³Academy of Military Sciences, Beijing 100097, China. ⁴College of Science, National University of Defense Technology, Changsha 410073, China. ⁵School of Physical Science and Technology, Lanzhou University, Lanzhou 730000, China. ⁶Hefei National Laboratory for Physical Sciences at Microscale and Department of Modern Physics, University of Science and Technology of China, Hefei, Anhui 230026, China. ⁷Beijing Synchrotron Radiation Facility, Institute of High Energy Physics, Chinese Academy of Sciences, Beijing 100049, China. ⁸Faculty of Science and Technology, Hirosaki University, Bunkyo-cho, Hirosaki-shi, Aomori 036-8561, Japan. ⁹Precision Spectroscopy Division, Japan Synchrotron Radiation Research Institute, Sayo, Hyogo 679-5198, Japan. ¹⁰Institute of Atomic and Molecular Physics, Jilin University, Changchun, Jilin 130012, China. ¹¹University of Würzburg, Institute of Theoretical Physics and Astrophysics, Am Hubland, 97074 Würzburg, Germany. ¹²These authors contributed equally: Ziyang Yuan, Hongxia Wang. ✉e-mail: adriana.palffy-buss@uni-wuerzburg.de; kongxiangjin@fudan.edu.cn

x-ray regime[5,13]. Nuclear resonances have been driven at both synchrotron and x-ray free electron laser (XFEL) sources. In both cases, x-ray intensities are measured, thus losing all information on the phase of the outgoing electromagnetic field. Also, without prior knowledge of the real or imaginary components of the nuclear resonance response, the Kramers-Kronig relations cannot be applied to extract these properties.

Synchrotron radiation (SR) has high brilliance and can be well focused on micrometre-size samples. Typically, SR is used in time-resolved nuclear forward scattering, removing the prompt off-resonant response by time gating[14]. In order to obtain intensity spectra in the energy domain, a Synchrotron Mössbauer Source can be used[15], which is however at present available only at few beamlines. An alternative and generally accessible technique employs time-resolved spectroscopy with an additional single-line reference sample, also known as analyzer, which is mounted on a Mössbauer (Doppler) drive and scans the nuclear transition frequencies by varying the velocity. Delayed forward scattered photons are recorded and integrated over time as a function of the detuning of the analyzer[16,17]. This time-integrated spectroscopy (TIS) method recovers the energy spectrum of the sample under investigation and was used in a number of recent experiments[18–22]. However, its accuracy heavily relies on the integration window (imposed by the beamline working parameters) and the analyzer thickness. Using a periodic time window, stroboscopic detection has been used as alternative to time integration[23–25]. In a different approach, a phase determination in the time domain (PDTD) algorithm using the maximum likelihood estimation has been developed[26]. A limiting requirement for PDTD is that the radiative coupling between the analyzer and the target is negligible. In addition, also PDTD is sensitive to the thickness of the used analyzer. Using this method, the energy spectrum could be obtained by a Fourier transformation using a truncated time window, at the expense of the energy resolution. Two recent experiments have extracted phase information based on physical models for this scheme, where an evolutionary algorithm has been performed using a Bayesian log-likelihood method[13,21]. Also, a method based on a rapidly oscillating reference sample has been proposed to measure both the amplitude and the phase of the spectral response[27].

Here, we demonstrate a method to recover both the amplitude and the phase of the electromagnetic field scattered off an unknown sample containing Mössbauer nuclei in the setup using an analyzer on a Doppler drive. Our nuclear phase retrieval spectroscopy (NPRS) method uses as input a full time- and energy-resolved data set provided by recording simultaneously the time of arrival and the corresponding Doppler velocity of the Mössbauer drive for each x-ray photon count. Depending on the choice of algorithm, a measured time spectrum of the analyzer alone can be used to increase the accuracy of the retrieved energy spectrum. The retrieval occurs without any theoretical modeling of the sample itself, delivering robust and accurate spectra. Since they are model-independent and data-based, all NPRS algorithms could be integrated in the data acquisition and analysis tools directly at SR and XFEL facilities.

## Results

### Experimental setup

The experimental setup and the NPRS input and output sets are illustrated in Fig. 1. A target containing $^{57}Fe$ Mössbauer nuclei with the first excited state at energy 14.4 keV and width $\Gamma_0 = 4.6$ neV is probed by a resonant but spectrally broad SR pulse with linear polarization. An analyzer containing the same nuclei is mounted on a Mössbauer drive that provides a periodic energy detuning. The motion of the Doppler drive induces an energy shift for the analyzer transmission function, with the magnitude of the shift determined by the drive's velocity. Both sample and analyzer contain enriched $^{57}Fe$. For the sample we use in the experiment $\alpha$ iron which presents hyperfine splitting of the

ground and excited nuclear states according to their spins $I_g = 1/2$ and $I_e = 3/2$. The sample magnetization is oriented perpendicular to the propagation and polarization directions of the input x-ray pulse. Due to selection rules, in this geometry the SR should drive only the two $\Delta m = m_e - m_g = 0$ transitions, where $m_{e(g)}$ are the nuclear excited (ground) state spin projections on the quantization axis. Single x-ray photons are detected by fast avalanche photodiode detectors (APDs)[28]. The 8 element APDs record the counts of the photons as a function of time and detuning. The delayed x-ray photons may arrive at the detector via three possible paths: scattered by the sample under investigation, scattered by the analyzer, or scattered by both sample and analyzer. The input data set is used by our NPRS method over several iterations to reconstruct the complex energy-dependent response of the sample.

### Algorithms

We develop three different mathematical algorithms for the phase retrieval procedure, all of them providing robust and accurate phase spectra. The most accurate energy spectra are provided by an algorithm using as input the 2D data set together with a measured response function of the analyzer alone and a theoretical model of the analyzer, still without modelling of the unknown sample. The core of this algorithm is a gradient-based phase retrieval reminiscent of methods used to solve the phase problem in imaging[3,29,30]. In the following we will refer to this version simply as NPRS algorithm. For the case a good theoretical model for the analyzer is not available, we present an alternative algorithm which uses only the input 2D data set and the measured time spectrum of the analyzer alone as constraint to recover the complex response functions of both sample and analyzer. We refer to this algorithm as a constrained blind version of NPRS (CB-NPRS), which is completely free of any theoretical modelling for sample or analyzer. A third algorithm labeled as blind NPRS (B-NPRS) considers the case that neither model nor measured spectrum of analyzer are known. Both B-NPRS and CB-NPRS algorithms have as starting point the Douglas-Rachford method known from ptychography[31–36].

The advantages of the three NPRS algorithms are two-fold. First, since they can actively extract information from all counts over the entire temporal spectrum at every velocity value, they recover accurate energy spectra. Here NPRS can provide finer details than the blind B-NPRS and CB-NPRS versions. However, comparisons for several setup examples show that all NPRS algorithms can provide more accurate energy spectra than TIS and PDTD. Second and most importantly, the NPRS algorithms provide reliable phase information for the scattered field without introducing any physical model for the sample, or in the case of B-NPRS and CB-NPRS not even for the analyzer. If required, the response function of the analyzer can be measured separately prior to the experiment and used for NPRS or CB-NPRS.

### Mathematical modeling

The starting point of all three NPRS algorithms is the time- and energy-dependent (measured) intensity $I(t, \Delta_D)$, where $\Delta_D$ is the Doppler detuning which shifts the value of the frequency seen by the analyzer foil moving with velocity $v$ as $\omega' = \omega(1 + v/c) = \omega + \Delta_D$. Here, $c$ stands for the speed of light. We let $R(\Delta)$ denote the response function of the sample under investigation and $T(\Delta - \Delta_D)$ the analyzer foil transmission, respectively, where $\Delta$ is the x-ray photon frequency. The phase to be retrieved refers to the phase of the complex sample response function $R(\Delta)$. The mathematical modeling of the scattered intensity in the setup of Fig. 1 is given by

$$I(t, \Delta_D) = \left| \frac{1}{\sqrt{2\pi}} \int_{-\infty}^{\infty} R(\Delta)T(\Delta - \Delta_D)e^{-i\Delta t}d\Delta \right|^2. \quad (1)$$

The component of the input pulse is not explicitly visible in the expression above, since we consider a normalized constant incoming

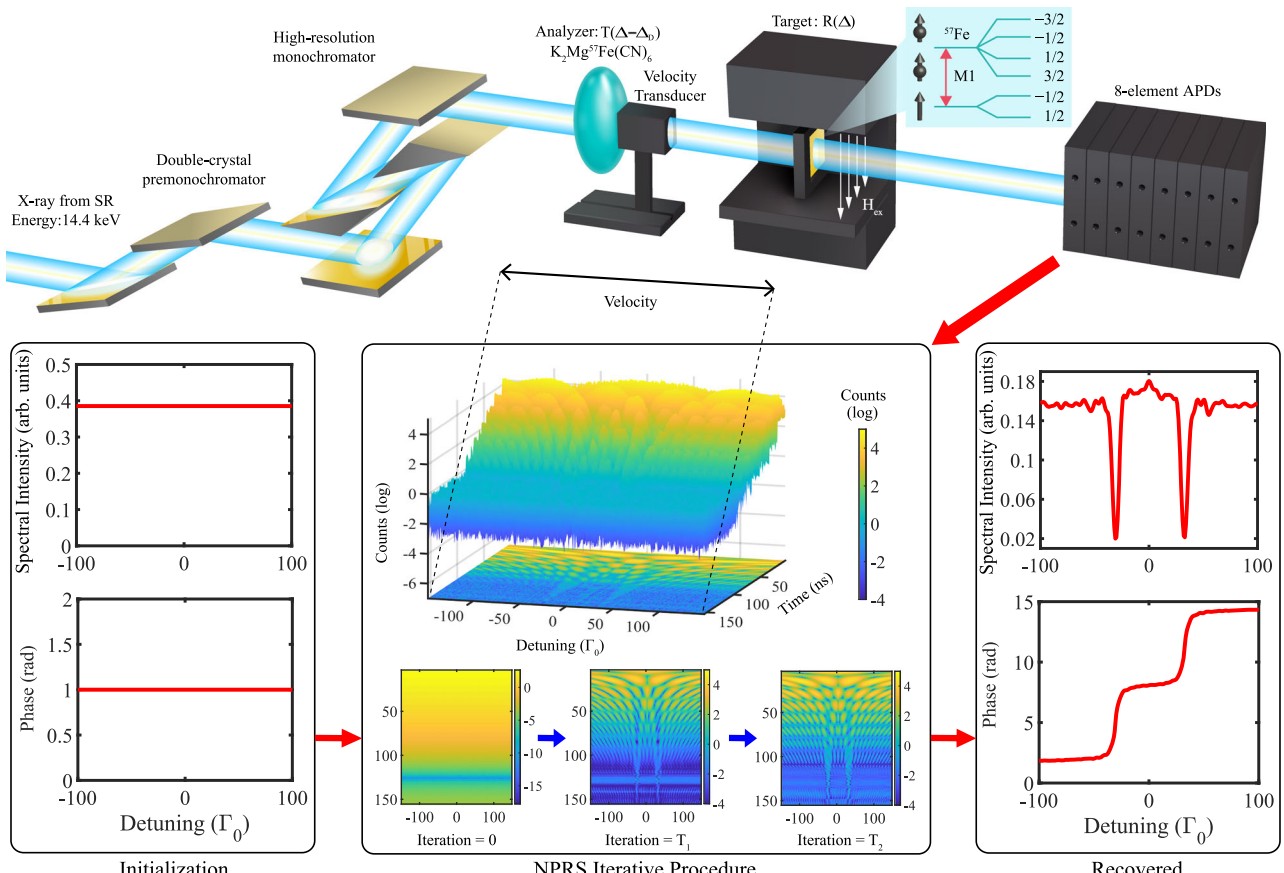

**Fig. 1 | Experimental setup and nuclear phase retrieval spectroscopy (NPRS) algorithms' sketch.** Resonant monochromatized synchrotron radiation (SR) passes through the analyzer with transmission function $T(\Delta - \Delta_D)$ mounted on the Mössbauer drive. The x-rays are then scattered off a $^{57}$Fe sample with response function $R(\Delta)$. Single x-ray photons are detected by avalanche photodiods (APDs) as a function of time and Doppler detuning. Our iterative NPRS algorithms are applied to recover the spectrum (intensity and phase) from these registered counts. Starting from a flat initialization, the algorithms iteratively improve the spectrum up to the desired precision. Here the example of a magnetized $^{57}$Fe sample with the SR pulse driving two nuclear hyperfine transitions is presented.

field $E_{in}(\Delta) = 1$ corresponding to the $\delta(t)$ pulse in the time domain. During the experiment, the photon counts $I(t_k, \Delta_D^l)$ are measured at $t_k \in \mathbf{E} = \{t_1, t_2, \cdots, t_K\}$ under different Doppler detunings $\Delta_D^l \in \mathbf{\Omega} = \{\Delta_D^1, \Delta_D^2, \cdots, \Delta_D^l\}$. If we discretize the energy range $[-\Delta_{max}, \Delta_{max}]$ by equally spaced nodes $\Delta_j$ with stepsize $\Delta_{step}$, and define the vectors $\mathbf{R} = (R(\Delta_0), R(\Delta_1), \cdots, R(\Delta_{n-1}))$, $\mathbf{T}_l = (T(\Delta_0 - \Delta_D^l), T(\Delta_1 - \Delta_D^l), \cdots, T(\Delta_{n-1} - \Delta_D^l))$, a discretized model can be formulated as

$$I(t_k, \Delta_D^l) = |\mathbf{F}_{:,k}^H(\mathbf{R} \odot \mathbf{T}_l)|^2 + \varepsilon, \quad t_k \in \mathbf{E}, \Delta_D^l \in \mathbf{\Omega}, \quad (2)$$

where $\mathbf{F}_{:,k} = \frac{\Delta_{step}}{\sqrt{2\pi}}(1, e^{i\Delta_{step} \cdot t_k}, \cdots, e^{i(n-1)\Delta_{step} \cdot t_k})$, $\odot$ is the Hadamard product, $(\cdot)^H$ represents the conjugate transpose, and $\varepsilon$ is the error caused by noise and discretization (see Supplementary Methods for details).

The NPRS algorithm takes as input $I(t, \Delta_D)$ and $T(\Delta - \Delta_D)$ and retrieves the complex response function $R(\Delta)$ of the unknown target. Since the complex analyzer transmission function is not accessible in experiments, a theoretical model for the analyzer is required. We abbreviate $I(t_k, \Delta_D^l)$ to $I(k, l)$ which denotes the $(k, l)$th element of the matrix $\mathbf{I}$, and then estimate $\mathbf{R}$ from $\mathbf{I}$ by the Bayesian method, which maximizes the log of the posterior conditional probability $P(\mathbf{R}|\mathbf{I})$. According to Bayes' rule, we have $P(\mathbf{R}|\mathbf{I}) \propto \overset{(a)}{P(\mathbf{I}|\mathbf{R})} \overset{(b)}{P(\mathbf{R})}$, where (a) is the likelihood which usually satisfies the Poisson distribution, and (b) allows to introduce the prior knowledge of $\mathbf{R}$. In practice, the physical model

for $\mathbf{R}$ is not always available. Here, we refrain from making use of (b) and consider a more general case without using any prior information of $\mathbf{R}$. The optimization model to recover $\mathbf{R}$ from $\mathbf{I}$ is established as

$$\underset{\mathbf{R} \in \mathbb{C}^n}{\text{minimize}} \, \ell(\mathbf{R}) = \sum_{k=1}^{K} \sum_{l=1}^{L} \left( \left|\mathbf{F}_{:,k}^H(\mathbf{R} \odot \mathbf{T}_l)\right|^2 - I(k, l) \log \left( \left|\mathbf{F}_{:,k}^H(\mathbf{R} \odot \mathbf{T}_l)\right|^2 \right) \right). \quad (3)$$

The expression above is a challenging non-convex optimization problem since $\ell(\mathbf{R})$ is nonconvex and thus many local minima may exist[37,38]. We develop a gradient-based algorithm combined with momentum restart and adaptive reweighted modules, which recovers both the amplitude and the phase of the response function (see Methods and Supplementary Methods for details).

When the analyzer transmission $T(\Delta-\Delta_D)$ is partially or completely unknown, for instance, when only the measured time spectrum is available, or when no information on the analyzer exists, the optimization problem (3) becomes more ill-posed. In such cases, gradient-based methods are empirically prone to stagnating at unsatisfactory points. In the field of phase retrieval, the Douglas-Rachford method[39], as a type of reflection algorithm, usually outperforms gradient-based methods[40]. Consequently, we reformulate (3) into a feasible problem and develop the CB-NPRS and B-NPRS algorithms based on the Douglas-Rachford method to recover both the amplitude and phase of the response function, as well as the analyzer, in the two scenarios mentioned above (see Methods and Supplementary Methods for further details).

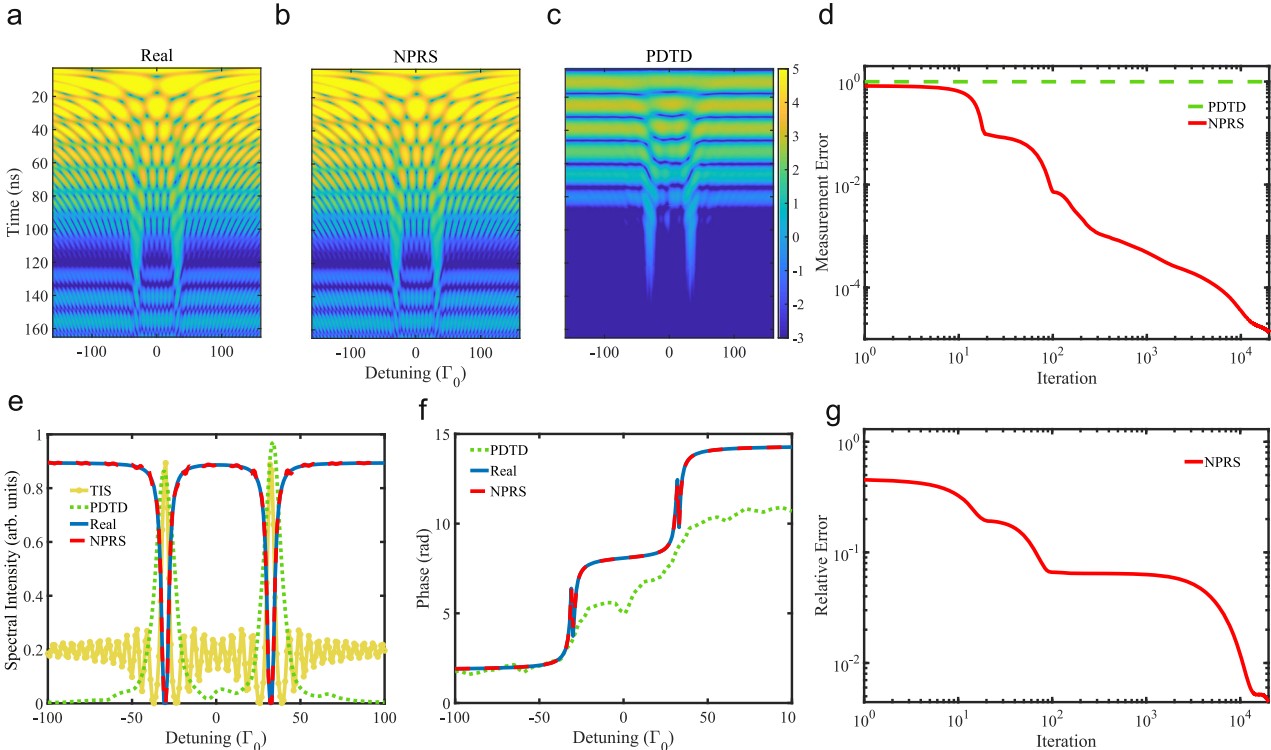

**Fig. 2 | Nuclear Phase Retrieval Spectroscopy (NPRS) algorithm applied to numerically simulated data for a nuclear forward scattering geometry.** We consider an $\alpha$-$^{57}$Fe sample of 2.3 $\mu$m with an aligned magnetization and an analyzer of 1 $\mu$m effective thickness. **a** Input 2D time- and energy-dependent intensity data set. The color code displays the natural logarithm of the signal. This set is then reconstructed via **b** NPRS and **c** phase determination in the time domain (PDTD).

**d** Measurement error of NPRS as a function of iteration step. **e** NPRS spectral intensity of the sample as a function of detuning. For comparison, we present here also the PDTD and scaled time-integrated spectroscopy (TIS) spectra in arbitrary units. **f** Recovered phase using NPRS and PDTD. **g** Relative error of the recovered spectrum using NPRS as a function of the retrieval iteration step.

## Numerical simulations

We have first tested the three NPRS algorithms using simulated data for four experimental scenarios of resonant nuclear x-ray scattering demonstrated in the literature: normal incidence targets containing $\alpha$-$^{57}$Fe with aligned and random magnetization orientations, forward scattering combined with fast mechanical motion as in the setup demonstrated in ref. 21, and thin-film cavities addressed in x-ray grazing incidence as in ref. 18. A thorough discussion of the potential and applicability of the NPRS algorithm and comparisons with other phase or spectrum retrieval methods is presented in this Section for the case of nuclear forward scattering with aligned magnetization introduced above, for which experimental results are addressed in the next Section. We focus first on the NPRS algorithm since a fairly accurate model for the analyzer used in the experiment is available, providing a comparison with the blind versions B-NPRS and CB-NPRS later on in the Experimental Results Section, while results for all algorithms and all considered setups are included in the Supplementary Methods.

We consider a linearly polarized x-ray pulse irradiating an $\alpha$-$^{57}$Fe target of (effective) thickness $d = 2.3$ $\mu$m in normal incidence as illustrated in Fig. 1. The simulated response function $R(\Delta)$ and analyzer transmission function $T(\Delta - \Delta_D)$ are obtained using a numerical implementation of the analytical expressions known for nuclear forward scattering[41]. We consider a single-resonance $K_2Mg^{57}Fe(CN)_6$ analyzer with the effective thickness $d \approx 1$ $\mu$m which coincides to what has been used in our experiment. We then use Eq. (1) to generate the 2D transmitted intensity input data set, shown here in Fig. 2a.

In order to simulate the experimental conditions at the SR beamline, the time range is set from 3 ns to 165 ns. The signals before 3 ns are excluded since the prompt off-resonant component of the incident x-ray pulse is very strong. The upper limit 165 ns corresponds to the bunch separation of the E mode at the nuclear resonant

scattering beamline BL09XU of SPring-8 in Japan, where the experiment discussed in the next Section was performed. The NPRS algorithm is applied to the truncated data and delivers the transmission $|R(\Delta)|^2$ and phase $\arg[R(\Delta)]$ spectra as function of detuning, shown in Figs. 2e,f, respectively. Out of these, we can also reconstruct the 2D intensity data set, which is presented in Fig. 2b.

Figures 2 e,f show for comparison the original simulated response function $R(\Delta)$ as transmission and phase. The retrieved and original simulated spectra display an excellent agreement; the algorithm recovers even the sharp phase jumps at approx. $\pm 30\Gamma_0$. To quantify this agreement, we define here the relative error as

$$\sqrt{\frac{\sum_{\Delta \in \mathbf{O}} |\hat{\mathbf{R}}(\Delta)e^{i\theta} - \mathbf{R}(\Delta)|^2}{\sum_{\Delta \in \mathbf{O}} |\mathbf{R}(\Delta)|^2}}, \tag{4}$$

where $\theta \in \arg\min_{\theta \in (0, 2\pi)} |\hat{\mathbf{R}}(\Delta)e^{i\theta} - \mathbf{R}(\Delta)|^2$, **O** is the region of interest, $\hat{\mathbf{R}}$ is the estimation (the retrieved value), and **R** is the ground truth (the input value from the numerical simulation, which is available in this case). The relative error as a function of the NPRS algorithm iteration is presented in Fig. 2g. With increasing number of iterations, the relative error drops to $10^{-2}$. This number quantifies the excellent agreement visible in Figs. 2e,f.

A more general type of error is the so-called measurement error, defined as

$$\sqrt{\frac{\sum_{k=1}^{K} \sum_{l=1}^{L} \left(I(k,l) - |\mathbf{F}_{:,k}^{\mathrm{H}}(\hat{\mathbf{R}} \odot \mathbf{T}_l)|^2\right)^2}{\sum_{k=1}^{K} \sum_{l=1}^{L} I^2(k,l)}}. \tag{5}$$

The measurement error provides means to quantify the agreement between input and retrieved data in an experiment, where only intensity counts are available. For the present case, the measurement error quantifies the comparison between the 2D spectra in Fig. 2a and b. The result is presented in Fig. 2d, showing that the measurement error drops down to $10^{-4}$ with increasing number of NPRS algorithm iterations.

At this point we can also compare the performance of the NPRS algorithm to other methods used in resonant nuclear x-ray scattering such as TIS or PDTD. Starting from the same 2D data set (obtained for a $1\,\mu m$-thickness analyzer), we reconstruct the spectral intensity (phase) via TIS and PDTD (just PDTD). For TIS, the retrieved spectra are very sensitive to the integration window and the thickness of the analyzer. The analyzer thickness for TIS is fixed by the $1\,\mu m$-value used for generating the 2D data set. We then optimize the integration range to obtain the best TIS fit of the original data. PDTD obtains the phase of the nuclear sample by a three-parameter fit of the 2D intensity data set. Once the complex time-dependent scattered field is retrieved, a Fourier transform can deliver the phase and intensity spectra as a function of energy, and the 2D spectrum can be reconstructed. Also the accuracy of PDTD is dependent on the thickness of the analyzer. We plot the obtained intensity and phase spectra (where available) next to the NPRS results in Fig. 2e and f. Both TIS and PDTD miss the prompt $\delta(t)$-like non-resonant scattering and therefore do not access the full response function of the sample. Nevertheless, recovered nuclear forward scattering energy spectra present resonant peaks that contain relevant information about the sample. Thus, in the energy-dependent intensity spectrum, NPRS presents two dips, while TIS and PDTD display two peaks at the same energies.

PDTD provides after several processing steps access to the phase introduced by the target. The numerical results of the PDTD method applied to the input 2D data set are presented in Fig. 2f and are compared to the exact result and the NPRS retrieved phase. While the NPRS phase is in perfect agreement with the original correct phase, the PDTD phase provides a reasonable approximation only for negative detunings in the interval $[-100\Gamma_0, -30\Gamma_0]$. Using the phase information, one can also use PDTD to reconstruct the 2D spectrum, which we present in Fig. 2c. While the main features of the 2D spectrum are still recognizable, most of the details are washed out. The calculated measurement error for the PDTD spectrum is given in Fig. 2d and is orders of magnitude larger than the NPRS error.

These comparisons demonstrate that the accuracy of the NPRS algorithm can be superior to the one of TIS and PDTD for this nuclear forward scattering setup. Further simulations and comparisons reach the same conclusion for normal incidence targets containing $\alpha$-$^{57}$Fe with different magnetization orientations, forward scattering combined with fast mechanical motion as in the setup demonstrated in ref. 21, and thin-film cavities addressed in x-ray grazing incidence as in ref. 18 (see Supplementary Methods and Supplementary Figs. 4-6). The NPRS measurement error remains below $10^{-4}$ for all four setups on a analyzer thickness range of $[0.5, 5]\,\mu m$, proving the robustness of NPRS (see Supplementary Fig. 12). This is in contrast to both TIS and PDTD whose performances are strongly dependent on the analyzer thickness, or even to Synchrotron Mössbauer Sources which also have limited resolution (see comparison in Supplementary Fig. 13). For simulated data sets which can provide sufficient number of points in the 2D spectrum, the NPRS algorithm can retrieve the spectrum in smallest detail, satisfying any practical purposes such as minute hyperfine splittings. However, as with all other methods, the NPRS algorithm is sensitive on the quality and range of the original 2D data set. The sensitivity to noise has been checked by using NPRS on 2D sets with superimposed Poisson noise with signal-to-noise ratios of 20, 45 and 60 dB. The results show a remarkable stability of the retrieved phase and a good recovery of the main features of the energy spectra (see Supplementary Figs. 7-10).

## Experimental results

Experiments were performed for the nuclear foward scattering setup discussed above at the nuclear resonant scattering beamline BL09XU of SPring-8 in Japan. The bunch mode provided a separation of 165.2 ns between the x-ray pulses (mode $E$) and a 2/29-filling bunch train which was blocked by the timing electronics. The pulse duration is determined by the length of the electron bunches in the storage ring and amounts to about 40 ps. To measure the 2D intensity data, an event-based data acquisition system was used to record temporal (time of arrival) and spectral (Doppler drive velocity) information for each signal photon separately. A single-line $K_2Mg^{57}Fe(CN)_6$ analyzer was used, whose complex spectral response $T(\Delta - \Delta_D)$ was determined from a measured time spectrum of the analyzer only (see Methods). The effective thickness of the analyzer was deduced to be approx. $d \approx 1\,\mu m$. A small external magnetic field was used to align the magnetization of the target such that only the two $\Delta m = 0$ transitions are driven.

The measured 2D time- and energy-dependent spectrum is shown in Fig. 3a. We have used the NPRS algorithm to retrieve the sample spectral intensity and phase, which are presented in Fig. 3e and f. The transmission spectrum of the $\alpha$-$^{57}$Fe sample presents the two dips already encountered in the previous Section for the numerical simulation, however here with more oscillations of the baseline. The recovered phase is also very similar to the numerical simulation result up to the sharp phase jumps which could not be recovered from the experimental 2D data set. Based on the retrieved sample response function, we have reconstructed the 2D data spectrum in Fig. 3b. In order to compare quantitatively the reconstructed and measured 2D spectra, we present the measurement error defined in Eq. (5) as a function of algorithm iteration in Fig. 3d. With increasing iterations, the measurement error drops below $10^{-1}$, demonstrating the good performance of the NPRS algorithm. The measurement error remains however much higher than the case of the numerical simulations, due to the lower quality of the input 2D spectrum.

For further checks, an independent time spectrum of the sample alone was also measured during the experiment. This gave us the opportunity to obtain a fit of the response function based on a theoretical model of the sample. From the time spectrum fit presented in Fig. 3g, we could deduce the sample thickness 2.289 $\mu m$, internal magnetic field $B = 32.537$ T and magnetic texture coefficient $0.985^{42}$. This allowed us to calculate the theoretical phase, and the time- and energy-dependent intensity spectra of the sample. The theoretical 2D spectrum thus obtained is presented in Fig. 3c. The measurement error using the experimental intensity values and the reconstructed sample response based on the theoretical model fit is shown in Fig. 3d. We see that for increased number of NPRS algorithm iterations, the measurement error for the retrieval is even smaller than the theoretical one, which in this case is however a consequence of overfitting the experimental noise. The time-dependent spectrum retrieved by NPRS is in excellent agreement with the measured data points (and their fit) as shown in Fig. 3g.

A comparison between the NPRS algorithm results and the simulated theoretical intensity and phase spectra based on the fitted sample parameters is presented in Figs. 3e,f, respectively. While the phases are in good agreement, up to the sharp phase jumps, the theoretical intensity displays (due to the fitted magnetic texture parameter not being equal to unity) four additional small peaks corresponding to the $\Delta m = \pm 1$ transitions between hyperfine levels. Thus, the fit of the time-domain spectrum of the target alone indicates that the magnetization geometry of the target is only approximately matching the originally planned driving of the $\Delta m = 0$ transitions only. NPRS recovers just the larger two of the four additional peaks from the experimental 2D spectrum, missing the other two minuscule ones. Three aspects should be mentioned here when interpreting these findings. First, the 2D data set has inferior count statistics and

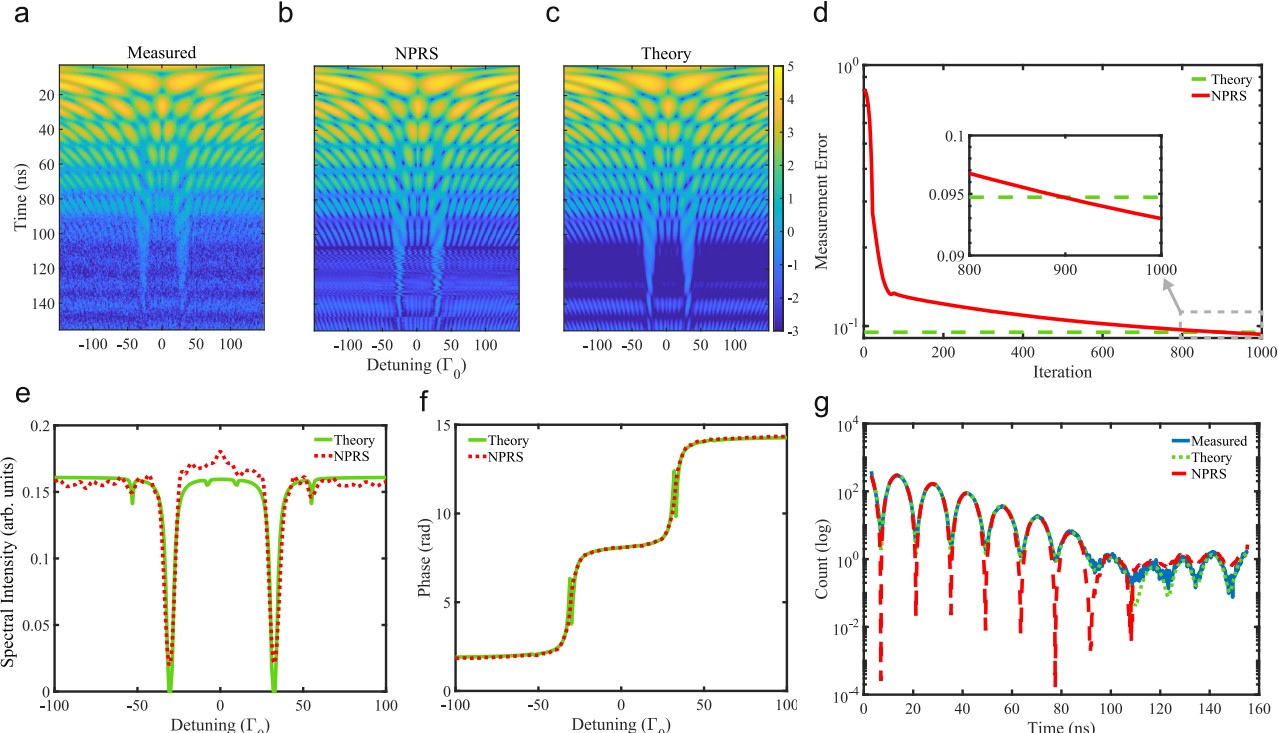

**Fig. 3 | Experimental and reconstructed results. a** Measured 2D intensity spectrum. The data points were binned, and the color coding indicates the function $\log_e(\#\text{ counts})$. Reconstructed 2D spectra using **b** the nuclear phase retrieval spectroscopy (NPRS) algorithm and **c** the theoretical model based on sample parameters fitted from an independently measured time spectrum. **d** Measurement errors of NPRS and theoretical model. **e, f** demonstrate the recovered spectra with

intensity and phase as a function of detuning. The NPRS spectra (red dashed line) are compared with the results of the theoretical model (green solid line). **g** Independently measured time spectrum of the sample only, theoretical model fit, and NPRS spectrum generated by a numerical Fourier transform from the recovered amplitude and phase.

additional noise due to the Doppler drive compared to the time spectrum data set. We have checked that NPRS would recover the full six-peak spectrum from a simulated 2D data set with better count statistics generated using the fitted sample parameters. Second, the theoretical response function in Fig. 3e is not retrieved directly from the experimental data. It is the plot of an analytical function obtained using a nuclear forward scattering theoretical model for the sample and fitted parameters, with no limitations in resolution. Third, the NPRS algorithm uses no knowledge of the sample model—which could in principle be added (when available) to increase the accuracy of the retrieved spectra.

We now turn to the results retrieved via the two algorithms B-NPRS and CB-NPRS which do not involve any analyzer model. Figure 4a and b shows the retrieved spectral intensities and phases, comparing NPRS with B-NPRS and CB-NPRS. We notice that all three methods are very robust in retrieving the phase of the sample response $\arg[R(\Delta)]$. The spectral intensities retrieved via the blind methods show larger baseline oscillations compared to NPRS. For the considered experimental sample, such oscillations render more difficult the identification of the small features induced by the non-unity value of the sample texture coefficient. Surprisingly, the measurement errors are dropping below the theoretical values, see Fig. 4c, with B-NPRS being the first to do so, despite the visible baseline oscillations in the recovered energy spectrum. This phenomenon is attributed to overfitting. Since real measurements contain noise and bias, methods with fewer constraints, specifically those that do not incorporate as much information about the analyzer, tend to fit these errors.

In Figs. 4d-f we compare the independently taken time spectrum of the sample alone with the calculated time spectra using the retrieved sample response for NPRS, CB-NPRS and B-NPRS. Here, the constrained blind version CB-NPRS has a better performance than B-

NPRS, especially for the late-time range. Overall our results show that both the blind and the blind constrained versions of NPRS retrieve the main features from the experimental 2D data set; especially, the retrieved phase is equally accurate for all algorithm versions and is superior to PDTD results (see also Supplementary Methods for further comparisons).

## Discussion

For a nuclear forward scattering setup, the NPRS algorithm can accurately retrieve phase and energy spectra from experimental data. Further results for 2D experimental spectra measured in grazing incidence geometry are presented in Supplementary Fig. 18 and discussed in the Supplementary Methods. Together with the retrieved spectra from numerically simulated 2D data sets for various resonant nuclear x-ray scattering setups, these results demonstrate the strength of NPRS for retrieval of reliable phase and energy spectra. Its particular advantages are that accurate results can be achieved without any physical model input for the sample response function and without being limited in resolution by the analyzer thickness.

The NPRS algorithm delivers a superior performance compared to the traditional TIS and PDTD methods. Compared to TIS, NPRS is more accurate, provides more information, is independent of integration limits and less sensitive to the analyzer thickness. Compared to PDTD, NPRS is demonstrably more accurate in phase retrieval and 2D spectrum reconstruction. In addition to these methods which have tradition for resonant nuclear x-ray scattering, one may consider also ptychography[43] as a candidate for obtaining the complex response function of the sample. To the best of our knowledge, this has not been performed so far for nuclear scattering. The conceptual similarities between the two methods allowed us to implement and test several ptychography algorithms for the four different setups of nuclear x-ray

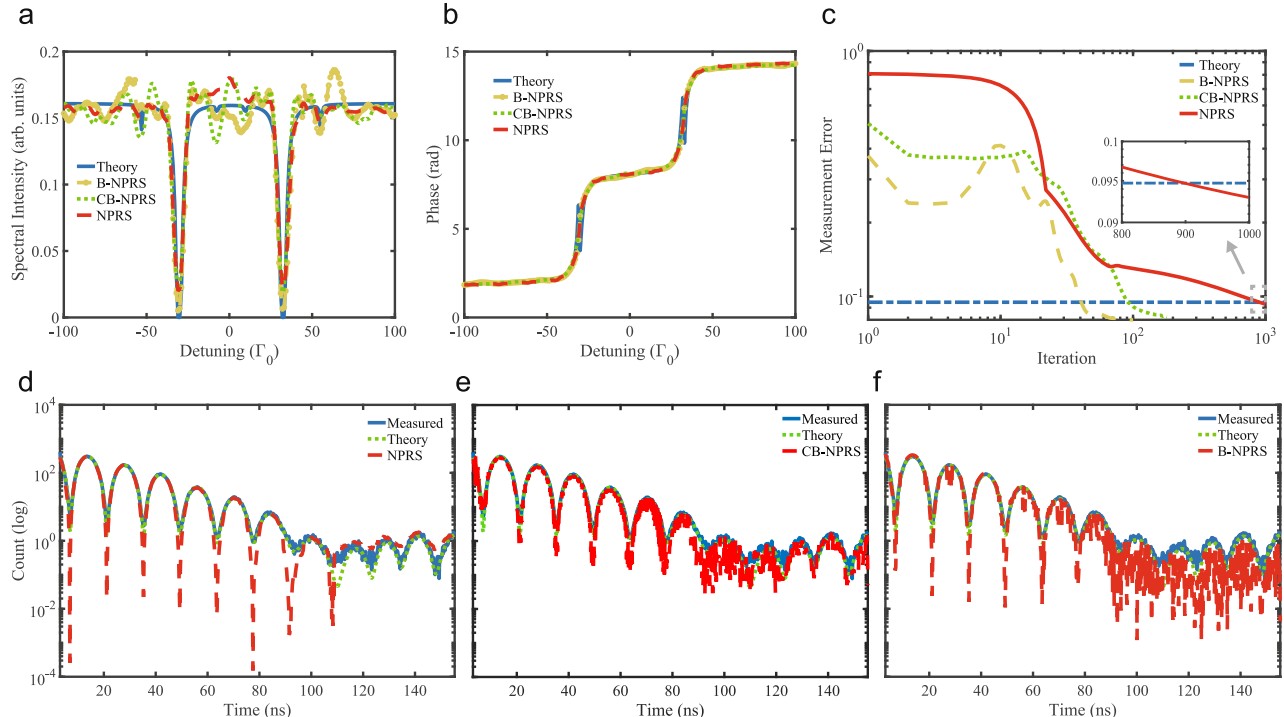

**Fig. 4 | Blind algorithms applied to experimental data set. a, b** The recovered spectra with intensity and phase as a function of detuning. The blind nuclear phase retrieval spectroscopy (B-NPRS, yellow line) and constrained blind nuclear phase retrieval spectroscopy (CB-NPRS, green line) spectra are compared with the results of the theoretical model (blue line) and the NPRS algorithm (red line). **c** Measurement errors of B-NPRS, CB-NPRS, NPRS and theoretical model. **d–f** Independently measured time spectrum of the sample only, theoretical model fit, and the spectrum generated by a numerical Fourier transform from the recovered amplitude and phase using **d** NPRS, **e** CB-NPRS and **f** B-NPRS.

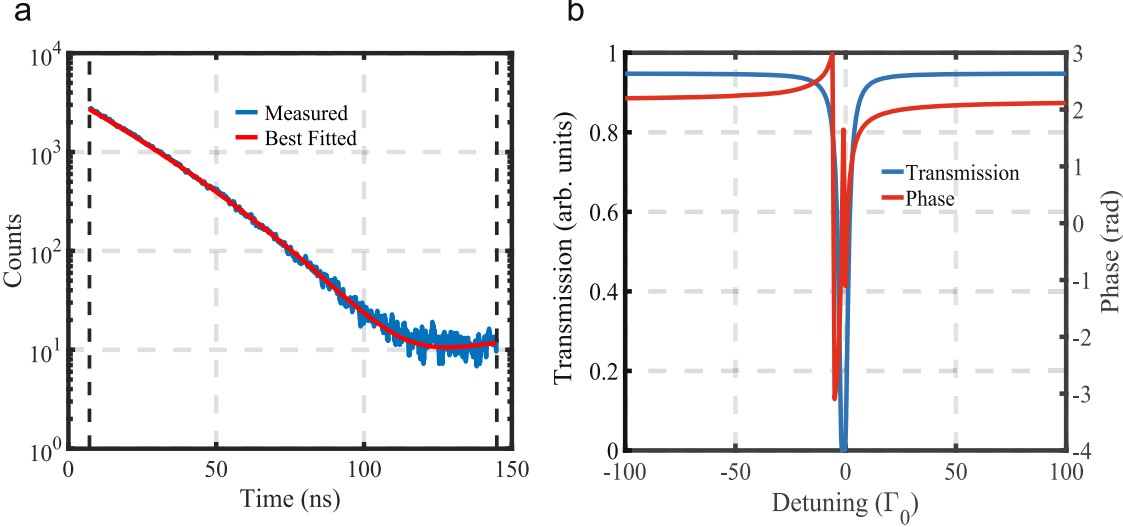

**Fig. 5 | Analyzer response function. a** Fit of the analyzer time spectrum. **b** Transmission and phase of the fitted response function for $\Delta_D = 0$.

scattering in discussion (see Supplementary Figs. 14–17). The comparison of transmission and phase spectra demonstrates that the NPRS algorithm is more efficient and can obtain the sample response function with the lowest measurement and relative errors at a smaller computational time cost than several established ptychography methods. Given the conceptual similarities between the NPRS algorithm, ptychography and coherent modulation imaging phase retrieval[44], it is likely that the efficiency of the developed NPRS algorithm could benefit also these other approaches.

The conceptual link to ptychography has been demonstrated by implementing also the blind algorithm versions B-NPRS and CB-NPRS, which do not require any theoretical modeling, neither for the sample (this feature is shared also by NPRS), nor for the analyzer. Compared to the NPRS algorithm, the blind algorithms are more challenging since both ambiguities of the solution as well as the coupling between the variables complicate the retrieval. While the retrieved spectral intensities capture the main features of the spectra, baseline oscillations might obscure minuscule features

thereof. However, the quality of the retrieved phase is very stable, allowing for accurate results even in the absence of a good analyzer model. This feature is unique and an important asset for the B-NPRS and CB-NPRS methods.

While the resonant nuclear scattering energy spectrum of $^{57}$Fe can be in principle measured at a Synchrotron Mössbauer Source, the field phase is more difficult to access and relies on retrieval algorithms. The overall NPRS method comes timely, as with the recent advances in nuclear quantum optics, for instance coherent x-ray optical control of nuclear excitons[13], or the driving of nuclear transitions with the X-ray Free Electron Laser (XFEL)[6,45], phase-sensitive measurements are likely to gain in significance as was the case for quantum optical setups at optical frequencies[46,47]. Our NPRS algorithms have potential applications in x-ray metrology[6,21,48], exploring x-ray quantum phenomena[13,19,49], or investigating novel topological effects in the x-ray frequency range. As an accurate, efficient and model-free (as far as the sample is concerned) phase retrieval method, all NPRS algorithms could be directly integrated at x-ray beamlines at SR and XFEL facilities and have a significant impact as analysis tool on future nuclear resonant x-ray scattering experiments.

## Methods
### Algorithms
The core of the NPRS method is the vanilla gradient descent (GD) algorithm, which is one of the most popular optimization algorithms for non-convex problems[50] and by far the most widespread method used in deep learning[51]. By choosing a proper initialization $\mathbf{R}^{(0)}$, the $m$th iteration takes the form

$$\mathbf{R}^{(m)} = \mathbf{R}^{(m-1)} - \mu \sum_{k=1}^{K} \sum_{l=1}^{L} \left(1 - \frac{I(k,l)}{\left|\mathbf{F}_{:,k}^{H}(\mathbf{R}^{(m-1)} \odot \mathbf{T}_l)\right|^2}\right)(\mathbf{F}_{:,k} \odot \overline{\mathbf{T}}_l)(\mathbf{F}_{:,k} \odot \overline{\mathbf{T}}_l)^{H}\mathbf{R}^{(m-1)},$$

(6)

where $\mu$ is the step size. Furthermore, the momentum with flexible parameter restart[52] and the adaptive reweighed[53] techniques are utilized to accelerate convergence and improve performance. The iterations will not stop until $\ell(\mathbf{R}^{(m)})$ is below some given error bound (more details are shown in Supplementary Methods).

The core of our CB-NPRS and B-NPRS algorithms is the Douglas–Rachford (DR) method, which serves as a classical algorithm for solving feasible problems. Let us define $\mathbf{\Psi}_l^{(0)} = \mathbf{R}^{(0)} \odot (\mathbf{C}_l \mathbf{T}^{(0)})$, $l = 1, \cdots, L$, where $\mathbf{C}_l$ is a sampling matrix. Then the $m$th iteration of the DR method reads

$$\mathbf{\Psi}_l^{(m)} = \mathbf{\Psi}_l^{(m-1)} + \mathbb{P}_{\mathcal{A}}\left(2\mathbb{P}_{\mathcal{B}}(\mathbf{\Psi}_l^{(m-1)}) - \mathbf{\Psi}_l^{(m-1)}\right) - \mathbb{P}_{\mathcal{B}}(\mathbf{\Psi}_l^{(m-1)}), \quad l = 1, \cdots, L,$$

(7)

where $\mathbb{P}_{\mathcal{A}}(\cdot)$ and $\mathbb{P}_{\mathcal{B}}(\cdot)$ are projection operators, and $\mathcal{A}$ and $\mathcal{B}$ are two constraint sets. The last iterated values $\mathbf{\Psi}_l$, $l = 1, \cdots, L$ are achieved when the measurement error falls below a given threshold. Subsequently, the final estimates $\hat{\mathbf{R}}$ and $\hat{\mathbf{T}}$ are decoupled from $\hat{\mathbf{\Psi}}_l$, $l = 1, \cdots, L$ (more details are given in Supplementary Methods).

### Analyzer response function
In order to obtain the transmission function $T(\Delta - \Delta_D)$, we fit a measured time spectrum of the analyzer alone (see Fig. 5). The isomer shift of the analyzer had been previously measured to be −0.1 mm/s relative to $\alpha$-$^{57}$Fe. The obtained fitted response function is $T(\Delta - \Delta_D) = (-0.54 + 0.81i)e^{-\frac{4.16\Gamma_0 i}{\Delta - \Delta_D + 1.03\Gamma_0 + 0.5\Gamma_0 i}}$, where $\Gamma_0$ is the spontaneous decay rate of $^{57}$Fe. As an example, the transmission and phase of the analyzer for the case when $\Delta_D = 0$ is shown in Fig. 5b.

## Data availability
The experimental data sets used here are available on the Zenodo repository, see ref. 54.

## Code availability
The NPRS algorithm script together with input data files are available on the Zenodo repository, see ref. 54. Additional scripts are available from the corresponding authors upon request.

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

## Acknowledgements

X.K., W. X., Y.J.Z. and J.Y. thank the National Key Research and Development Program of China for funding this research via Grant No. 2024YFA1610900. Z.Y. and H.X.W. acknowledge support by the National Key Research and Development Program under Grant No. 2020YFA0713504 and by the National Natural Science Foundation of China (NSFC) under Grant No. 12471401. X.K. acknowledges support by NSFC under Grant No. 12447106, No. 12147101 and No. 11904404, and the 111 project. A.P. gratefully acknowledges support from the German Science Foundation (Deutsche Forschungsgemeinschaft, DFG) in the framework of the Heisenberg Program (Project PA 2508/3-1), Project No. 429529648 (TRR 306 QuCoLiMa) (Quantum Cooperativity of Light and Matter) and the Cluster of Excellence on Complexity and Topology in Quantum Matter - ct.qmat (EXC 2147, Project No. 390858490). J.Y. acknowledges the support by the National Key Research and Development Program of China under Grant No. 2019YFA0307700. Z.L. acknowledges support by NSFC under Grant No. 11704167. L.Z. acknowledges support by NSFC under Grant No. 12334010. W. X. and Y.J.Z. are supported by NSFC under Grant No. 12075273, and the high-energy photon source (HEPS), a major national science and technology infrastructure. We acknowledge the support of the nuclear resonant scattering beamline of SPring-8 in Japan for Proposals No. 2020A1101 and 2021B1522, and the allocation of beamtime by beamline 1W1A of Beijing Synchrotron Radiation Facility (BSRF). We would like to thank Dr. Jialu Wu, Prof. Jing Ma, Prof. Yuanhua Lin and Prof. Cewen Nan from Tsinghua University for support with the sample reflectivity measurements.

## Author contributions

X.K and A.P. proposed this project and devised the experimental concept. Z.Y. and H.X.W. developed the mathematical model and the algorithms. R.M. and Y.Y set up the beamline instrumentation and data acquisition systems and operated the beamline. Z.L., T.W., H.W., X.H., T.L. and Z.M. fabricated the samples. Z.Y. and X.K. performed the numerical simulations and the experimental data analysis. X.K., Z.Y., H.X.W. and A.P. wrote the manuscript. L.Z., W.X, Y.Z., Y.C. and J.Y. participated in discussing the experimental results and editing the manuscript.

## Funding

## Competing interests

The authors declare no competing interests.
