## [Transparent Peer Review file · Nature Communications]

Nuclear phase retrieval spectroscopy using resonant x-ray scattering

Corresponding Author: Professor Adriana Pálffy

Version 0:

Reviewer comments:

Reviewer #1

(Remarks to the Author)

In my previous report, I suggested publication, but restricted this recommendation by pointing out few remaining weaknesses of the manuscript. In their revised version, these have all been resolved, and I now recommend publication without any restrictions.

It is indeed a very difficult task to demonstrate superiority over previous schemes aiming at similar problems. Therefore I very much appreciate the authors' new approach of focusing on the model-free phase retrieval. This represents a significant step forward, strongly supported by the new numerical calculations involving noise and the additional corresponding analysis of the experimental data. Overall the manuscript now clearly demonstrates new and significant results which warrant a publication in Nature Communications. The present method indeed could evolve into the standard approach for such problems.

My other remarks have also been resolved. The relation to previous methods has adequately been clarified, and the missing information on the experiment and analysis has been added.

The supplementary information is very valuable, and provides all details of the methodology. It will allow the community to adopt the new method. In particular, Fig. S19 (also in the reply) clearly illustrates the relation between the different approaches. In practice, a convergence of the different approaches can be used to further strengthen the reliability of the analysis.

(Remarks on code availability)

I cannot test/review the code since I do not use Matlab. However, this is not a shortcoming of the provided code. Given that Matlab is widely used, the code should be useful to many researchers. It does contain instructions which appear suitable to me.

Reviewer #2

(Remarks to the Author)

The authors provided further explanation of their phase retrieval algorithm. I have a few questions and suggestions that could strengthen the manuscript:

1) In principle, the desired measurements are the real and imaginary parts of the susceptibility or refractive index. The real part of susceptibility leads to phase shifts, while its imaginary part causes absorption. It is well known that the Kramers–Kronig relation can determine the refractive index from detuning-dependent absorption. Explaining why your algorithm is superior to the Kramers–Kronig approach in the introduction would provide stronger support for the significance of your work.

2) I found the following statements of the manuscript appear inconsistent:

- a) Your description of the three possible paths to the detector (page 4, line 66).
- b) Equation (1).
- c) Your previous response to my question: "The x-rays scattered by the target have already been scattered via the analyzer. The phase retrieved is the phase imprinted by the target only, independently of the analyzer."
- d) The discussion on page 10, line 195, about the drawbacks of TIS and PDTD due to missing the prompt $\delta(t)$ -like input.

2.1) I do not see the prompt $\delta(t)$ term in your Eq. (1) either.

2.2) Should the prompt $\delta(t)$ input be described by a fourth path?

2.3) Your Eq. (1) seems to account for x-rays scattered by both the sample and analyzer. How do the other two paths factor into this equation?

2.4) If the x-ray field is modeled as $E(t) = \delta(t) + \text{path1}(t) + \text{path2}(t) + \text{path3}(t)$, why does Eq. (1) only include the cross-term of $\text{path3}(t)$?

Addressing above points in the manuscript would clarify the theoretical framework and strengthen the arguments.

3) Regarding Figure 1:

Do I understand correctly that:

The x-axis of the red curves represents Delta;

The x-axis of the middle 2-D spectra represents Delta_D, as indicated by the velocity?

Initially, it seemed counterintuitive that the x-ray phase shift from the target would be affected by the analyzer velocity. Could you clarify this distinction further in the manuscript?

I find your work interesting, but I am not fully convinced yet. I look forward to seeing how you address my questions and comments.

(Remarks on code availability)

Version 1:

Reviewer comments:

Reviewer #2

(Remarks to the Author)

The authors have addressed all my questions thoroughly. I recommend the manuscript for publication.

(Remarks on code availability)

Response to Report of Reviewer 1/NCOMMS-24-64787-T

1. *“In my previous report, I suggested publication, but restricted this recommendation by pointing out few remaining weaknesses of the manuscript. In their revised version, these have all been resolved, and I now recommend publication without any restrictions.*

It is indeed a very difficult task to demonstrate superiority over previous schemes aiming at similar problems. Therefore I very much appreciate the authors’ new approach of focusing on the model-free phase retrieval. This represents a significant step forward, strongly supported by the new numerical calculations involving noise and the additional corresponding analysis of the experimental data. Overall the manuscript now clearly demonstrates new and significant results which warrant a publication in Nature Communications. The present method indeed could evolve into the standard approach for such problems.

My other remarks have also been resolved. The relation to previous methods has adequately been clarified, and the missing information on the experiment and analysis has been added.

The supplementary information is very valuable, and provides all details of the methodology. It will allow the community to adopt the new method. In particular, Fig. S19 (also in the reply) clearly illustrates the relation between the different approaches. In practice, a convergence of the different approaches can be used to further strengthen the reliability of the analysis.”

“I cannot test/review the code since I do not use Matlab. However, this is not a shortcoming of the provided code. Given that Matlab is widely used, the code should be useful to many researchers. It does contain instructions which appear suitable to me.”

Reply - We thank Reviewer 1 for wholeheartedly recommending publication of our manuscript and for acknowledging the potential of our new model-free phase retrieval methods. We are pleased that the revision has successfully addressed all the points raised in Reviewer 1’s previous report. We also appreciate Reviewer 1’s positive comments regarding the new numerical calculations, experimental data analysis, and the potential impact of our method.

Response to Report of Reviewer 2/NCOMMS-24-64787-T

1. *“The authors provided further explanation of their phase retrieval algorithm. I have a few questions and suggestions that could strengthen the manuscript: 1) In principle, the desired measurements are the real and imaginary parts of the susceptibility or refractive index. The real part of susceptibility leads to phase shifts, while its imaginary part causes absorption. It is well known that the Kramers-Kronig relation can determine the refractive index from detuning-dependent absorption. Explaining why your algorithm is superior to the Kramers-Kronig approach in the introduction would provide stronger support for the significance of your work.”*

Reply - We thank the Reviewer for this suggestion. Our method focuses on retrieving the phase of the complex-valued response function of the target $R(\Delta)$, specifically the argument $\arg[R(\Delta)]$, as a function of detuning. In addition to phase retrieval, our algorithms also recover the spectral intensity as the absolute value squared of the response function, $|R(\Delta)|^2$. Unlike the Kramers-Kronig relations, which connect the real and imaginary parts of the response function, our method does not need any prior information of the real or imaginary parts of a response function to retrieve its phase. This is a clear advantage over the Kramers-Kronig approach.

Corresponding changes to the manuscript:

We have added the discussion about the advantage of our methods compared to Kramers-Kronig approach in the introduction (see text highlighted in blue on page 3 of the revised manuscript).

Also, without prior knowledge of the real or imaginary components of the nuclear resonance response, the Kramers-Kronig relations cannot be applied to extract these properties.

2. *“2) I found the following statements of the manuscript appear inconsistent:*
 - a) *Your description of the three possible paths to the detector (page 4, line 66).*
 - b) *Equation (1).*
 - c) *Your previous response to my question: “The x-rays scattered by the target have already been scattered via the analyzer. The phase retrieved is the phase imprinted by the target only, independently of the analyzer.”*
 - d) *The discussion on page 10, line 195, about the drawbacks of TIS and PDTD due to missing the prompt $\delta(t)$ -like input.*
 - 2.1) *I do not see the prompt $\delta(t)$ term in your Eq. (1) either.*
 - 2.2) *Should the prompt $\delta(t)$ input be described by a fourth path?*
 - 2.3) *Your Eq. (1) seems to account for x-rays scattered by both the sample and analyzer. How do the other two paths factor into this equation?*
 - 2.4) *If the x-ray field is modeled as $E(t) = \delta(t) + \text{path1}(t) + \text{path2}(t) + \text{path3}(t)$, why does Eq. (1) only include the cross-term of $\text{path3}(t)$?”*

“Addressing above points in the manuscript would clarify the theoretical framework and strengthen the arguments.”

Reply - We thank the Reviewer for this comment, which gives us the opportunity to clarify the connections between all the quoted statements. All statements in the manuscript and in our previous reply are consistent, as detailed in the following.

Let us consider first the experimental setup. A short synchrotron radiation pulse passes through the analyzer, then through the sample and finally reaches the detector. All photons reaching the detector must have passed via analyzer and sample; however, this does not necessarily mean that they have undergone nuclear scattering. The photons which passed the analyzer and sample without nuclear excitation are prompt. The delayed photons come much later, having being absorbed and reemitted by the iron nuclei, which have a lifetime of 141 ns. Thus, the delayed photons must have undergone at least one nuclear scattering, either in the analyzer, or in the sample, or both. This was the statement 2(a) quoted by the Reviewer, and those are the three possible paths of *delayed* photons arriving at the detector.

Let us now address the theoretical modelling of the measured scattered field intensity. There are two approaches to calculate the joint time response of the target and the analyzer, either in the frequency domain, or directly in the time domain. In the latter, the input pulse is approximated with a Dirac delta peak $\delta(t)$. This corresponds to a constant input pulse in the frequency domain.

1. Time-domain approach: This method calculates the total response in the time domain by convoluting the time responses of the analyzer and target with the $\delta(t)$ input. Note that the total response cannot be simply modeled as the sum of contributions from different paths as suggested in 2.4), as the response is determined through convolution rather than direct summation.

2. Frequency-domain approach: This involves calculating the joint frequency response of the target and analyzer, followed by a Fourier transform to obtain the time-domain response. This method is employed in our manuscript. Specifically, Eq. (1) in the main text is

$$I(t, \Delta_D) = \left| \frac{1}{\sqrt{2\pi}} \int_{-\infty}^{\infty} R(\Delta) T(\Delta - \Delta_D) e^{-i\Delta t} d\Delta \right|^2. \quad (1)$$

Here, $R(\Delta)$ and $T(\Delta - \Delta_D)$ are the frequency responses of the target and the analyzer, respectively. The frequency distribution of the input pulse, $E_{in}(\Delta)$, is not explicitly visible in Eq. 1 because we consider $E_{in}(\Delta) = 1$, corresponding to a $\delta(t)$ input in the time domain. Thus, the prompt $\delta(t)$ input is present and accounted for in the frequency domain in Eq. (1).

We would like to point out that Eq. (1) accounts for all possible paths, not only the double nuclear scattering as suggested in 2.3). Whether an x-ray photon is resonantly scattered or not in the analyzer is determined by the shape of the analyzer transmission $T(\Delta - \Delta_D)$; for instance, we have approx. 100% transmission for off-resonance energies, but strong absorption at the resonance energies. Likewise, the information about whether an x-ray photon is resonantly scattered in the sample is available in the sample response function $R(\Delta)$. Thus, Eq. (1) describes all possible cases: prompt photons which have not experienced any nuclear scattering, and delayed photons which have been scattered by nuclei in the analyzer, in the sample, or in both.

In conclusion, the statements in the manuscript are consistent. To avoid any confusion, we have added a sentence to explain Eq. (1) more explicitly in the revised manuscript.

Corresponding changes to the manuscript:

We have added a sentence to explain Eq. (1) in the main text more clearly, see text highlighted in blue on page 6 of the revised manuscript:

The component of the input pulse is not explicitly visible in the expression above, since we consider a normalized constant incoming field $E_{in}(\Delta) = 1$ corresponding to the $\delta(t)$ pulse in the time domain.

3. “Regarding Figure 1:

Do I understand correctly that:

The x-axis of the red curves represents Δ ;

The x-axis of the middle 2-D spectra represents Δ_D , as indicated by the velocity?

Initially, it seemed counterintuitive that the x-ray phase shift from the target would be affected by the analyzer velocity. Could you clarify this distinction further in the manuscript?”

Reply - We confirm that your understanding of the x-axes is correct. To clarify, the phase shift imprinted by the target is not affected by the analyzer velocity. The phase shift is determined by $R(\Delta)$, which remains independent of the analyzer velocity. The analyzer velocity introduces a Doppler shift, denoted by Δ_D , which modifies the response function $T(\Delta - \Delta_D)$. This is how the analyzer velocity influences the 2D spectrum $I(t, \Delta_D)$, as can be seen in Eq. (1). Our method retrieves both the phase and amplitude of the response function $R(\Delta)$ from the 2D spectrum $I(t, \Delta_D)$.

We have clarified this point by addressing more explicitly the Doppler shift in the manuscript.

Corresponding changes to the manuscript:

The experimental setup and the NPRS input and output sets are illustrated in Fig. 1. A target containing ^{57}Fe Mössbauer nuclei with the first excited state at energy 14.4 keV and width 4.6 neV is probed by a resonant but spectrally broad SR pulse with linear polarization. An analyzer containing the same nuclei is mounted on a Mössbauer drive that provides a periodic energy detuning. **The motion of the Doppler drive induces an energy shift for the analyzer transmission function, with the magnitude of the shift determined by the drive’s velocity.** Both sample and analyzer contain enriched ^{57}Fe .

4. “I find your work interesting, but I am not fully convinced yet. I look forward to seeing how you address my questions and comments.”

Reply - We thank the Reviewer for this rather positive feedback. We appreciate the interest in our work and have carefully addressed all questions and comments. Consequently, we are confident that the revisions clarify the points raised and strengthen the manuscript.